# Family planning uptake and its associated factors among women of reproductive age in Uganda: An insight from the Uganda Demographic and Health Survey 2016

Anthony Mark Ochen 1☯*, Che Chi Primus 2☯

1 Department of Health Services, Alebtong District Local Government, Lira, Uganda, 2 KEMRI-Wellcome Trust Research Programme, Kilifi, Kenya

☯ These authors contributed equally to this work.
* markochen@rocketmail.com

## Abstract

Despite the government efforts to reduce the high fertility levels and increase the uptake of family planning services in Uganda, family planning use was still low at 30% in 2020 which was the lowest in the East African region. This study was undertaken to determine the prevalence and factors associated with the uptake of family planning methods among women of reproductive age in Uganda. This community-based cross-sectional study utilized secondary data from the Uganda Demographic and Health Survey (UDHS) of 2016. The survey data was downloaded from the Measure Demographic Health Survey website after data use permission was granted. Data was collected from a representative sample of women of the reproductive age group (15–49 years) from all 15 regions in Uganda. A total of 19,088 eligible women were interviewed but interviews were completed with 18,506 women. Data analysis was performed using SPSS statistical software version 32.0 where univariable, bivariable, and multivariable analyses were conducted. The prevalence of family planning use was found to be 29.3% and that of modern contraceptive use was found to be 26.6%. Multivariable analysis showed higher odds of current family planning use among older women (40–44 years) (aOR = 2.09, 95% CI: 1.40–3.12); women who had attained the secondary level of education (aOR = 1.91, 95% CI: 1.32–2.76); those living in households with the highest wealth index (aOR = 1.87, 95% CI: 1.29–2.72); and awareness of the availability of family planning methods (aOR = 1.41, 95% CI: 1.17–1.72). In conclusion, the study suggests improving women's education attainment, socio-economic position, and awareness may help increase use in the population.

## Introduction

Reducing the number of unplanned pregnancies can be achieved through better birth spacing, as children born less than two years before or after the birth of their siblings have been found to have a higher rate of mortality during their first five years of life [1]. In addition, it will

**Data Availability Statement:** The dataset used is openly available upon permission from the MEASURE DHS website (URL. http://www.dhsprogram.com/data/available.datasets.cmf).

**Funding:** The authors received no specific funding for this work.

**Competing interests:** The authors have declared that no competing interests exist.

lower the number of infants born at extremely high mortality risk because their mothers died during or soon after delivery. Uganda has one of the fastest-growing populations in the sub-Saharan Africa (SSA) region at a rate of 3.2% per annum [2]. It has a persistently high fertility rate of 5.4 children born per woman which is higher than the total wanted fertility rate of 4.3 [3]. The use of family planning (FP) among women increased from 23% in 2000 to 39% in 2016, however, the increase was most pronounced for the use of modern methods which rose from 18% in 2001 to 35% in 2016 [3]. The total population of Uganda was 34.6 million persons in 2014 representing an average annual growth rate of 3.0% between 2002 and 2014 [4]. Uganda aspires to become a middle-income Country by 2025, however, the country has only managed a decline in poverty levels from 24.3% in 2010 to 19.7% in 2015 [5]. The 2002 health financing strategy estimated that for the sector to be able to provide the Uganda National Minimum Health Care Package, USD 28 per capita expenditure would be required. However, for the Financial Year 2013/14, only USD 12.0 per capita (which includes donor projects and Global Health Initiatives) was available [6]. The Total Fertility rate in Uganda declined from 7.1 children per woman in 1991 to 5.8 children per woman in 2014 [4].

The Uganda Demographic and Health Survey (UDHS) 2016 estimates that the total demand for family planning in Uganda among women increased from 58% in 2000–01 to 67% in 2016, and the proportion of demand satisfied by modern methods increased from 18% to 35% over the same period. The unmet need has decreased slightly since 2000, from 35% to 28% in 2016 [3]. Despite these slight gains, there were still 336 maternal deaths per 100,000 live births and an infant mortality rate of 43 deaths per 1,000 live births [3]. This poses a great threat to the development and well-being of the Ugandan population as reflected in the high infant mortality rates and maternal mortality ratios. High birth rates not only affect maternal and child mortality but also frustrate governments' efforts in the provision of social and health services to communities.

The World Health Organisation (WHO) refers to family planning (FP) as a process that allows people to attain their desired number of children and determine the spacing of pregnancies, which is achieved through the use of family planning methods and treatment of infertility [7]. Of the 1.9 billion women of reproductive age group (15–49 years) worldwide in 2019, 1.11 billion needed to space/cease future pregnancy; of these, 842 million used modern contraception, and 270 million had an unmet desire to space/cease future pregnancy [8]. The modern contraceptive prevalence among women of reproductive age increased worldwide between 2000 and 2019 by 2.1% from 55.0% to 57.1% [8]. Some of the sustained preferences for the large family size are a result of limited choice of methods; limited access to services particularly among young, poorer, and unmarried people; fear or experience of side effects; cultural or religious opposition; poor quality of available services; users and providers bias against some methods; and gender-based barriers to accessing services [8].

The United Nations (UN) estimates the total fertility rate (TFR) of the sub-Saharan Africa region at 4.7 births per woman in 2015–2020 which is more than twice the level of any other world region [9]. Consequentially, the population of sub-Saharan Africa is expected to grow from 1 billion in 2015 to about 2 billion in 2050 and nearly 4 billion in 2100 [9]. Therefore, family planning services are voluntary but access to the wide range of contraceptive methods for women to choose from may enhance their health prospects and have comprehensive benefits for their societies' social and economic development. There are great benefits to investing in family planning including reduced maternal and neonatal mortality through decline in abortions and pregnancies [10]. For this reason, numerous scholars have pointed out that promoting voluntary access to a wide variety of contraceptive methods for women is an important component of countries' strategies to advance social and economic development [11, 12]. This is well articulated in the Sustainable Development Goals (SDG) 3, target 3.7 calls on countries

"by 2030, to ensure universal access to sexual and reproductive health-care services, including for FP, information and education, and the integration of reproductive health into national strategies and programs"; with specifically 3.7.1 which calls for universal access to FP services to ensure healthy lives and well-being [13].

Despite the government efforts to reduce high fertility levels and increase uptake of FP services in Uganda, the prevalence rate was only 30% in 2020 among married women which was the lowest in the East African region [14]. The known factors contributing to the low use of family planning methods are multi-factorial and include; limited accessibility to contraceptives, long distance to the health facility, few qualified health experts, fear of side effects, limited male involvement, religion or cultural beliefs, polygamous marriage, and lack of awareness [15–20]. Monitoring factors influencing the uptake of FP services is important to target scarce public resources to those with more need and enhance the progress towards achieving the global targets. Family Planning is central to gender equality and women's empowerment and is a key driver of all 17 Sustainable Development Goals. Family planning saves lives, improves maternal and child health outcomes, and lifts families out of poverty by helping women have fewer children and freeing them to participate in the labor force [21]. Furthermore, family planning remains the low-cost, high-dividend investment option for addressing Uganda's high Total Fertility Rate (TFR), high school drop-out rates as a result of teenage pregnancy, and high Maternal Mortality Ratio (MMR), as well as improving the health and welfare of women and girls including families [21]. Similarly, the 2020 Demographic Dividend Report demonstrates that investing in family planning will accelerate fertility decline; coupled with mortality decline, the ratio of working-age adults would significantly increase relative to young dependents, thus propelling Uganda towards a middle-income country [22] Thus far, a recent study that has been published in Uganda only focussed on factors associated with modern contraceptives among female adolescents [23]. Therefore, this study was undertaken to examine the prevalence and factors associated with the current family planning uptake among women of reproductive age in Uganda using the 2016 Uganda Demographic and Health Survey data.

## Methods and materials

### Ethics statement

The survey was approved by the Uganda National Council for Science and Technology (UNCST). Respondents were informed about the survey and informed consent was obtained from participants. The authors received the survey data from the USAID DHS program database after a request to download the dataset was granted. After data access was authorized, the authors of this study maintained the confidentiality of the dataset [24].

### Study context

This study utilized secondary data from the Uganda Demographic and Health Survey (UDHS) 2016. The UDHS 2016 is a part of the global program implemented by the Uganda Bureau of Statistics (UBOS) in collaboration with the Ministry of Health (MoH). The funding for the UDHS 2016 was provided by the Government of Uganda, the United States Agency for International Development (USAID), the United Nations Children's Fund (UNICEF), and the United Nations Population Fund (UNFPA). The DHS is undertaken every five years and the 2016 survey is the sixth DHS in Uganda, the first one was conducted in 1988.

To generate statistics that were representative of the country as a whole in the 15 regions, the number of women surveyed in each region contributed to the size of the total sample in proportion to the population size of each region. This is because some regions had small populations and others had large populations. The 15 regions of Uganda where the UDHS 2016 was

## UGANDA

**Fig 1. Map of Uganda showing all the 15 regions.**

implemented were; South-Central, North-Central, Kampala, Busoga, Bukedi, Bugisu, Teso, Karamoja, Lango, Acholi, West-Nile, Bunyoro, Tooro, Kigezi, and Ankole Regions ""Fig 1,""

## Study design

This was a community-based cross-sectional study where data was collected from a representative sample of women of the reproductive age group (15–49 years). The data collection method was household surveys and women were interviewed at home.

## Target population

The study participants were all women in the reproductive age group of 15–49 years living in Uganda at the time of the survey.

## Sample size

A nationally representative sample of 20,880 households was selected for the study. From these households, a total of 19,088 eligible women in the reproductive age group were interviewed using a structured questionnaire [25]. However, interviews were completed with 18,506 women, yielding an overall response rate of 97%. Response rates were higher in rural (97.6%) than in urban areas (94.8%).

## Sampling procedure

The UDHS 2016 used a multi-stage stratified sampling method (in two stages) to select the study participants. Three regions (South Central, North Central, and Busoga) were stratified into island and non-island sub-regions. Each region/sub-region was stratified into urban and rural areas yielding 34 sampling strata. Samples of EAs were selected independently in each stratum in two stages. Implicit stratification and proportional allocation were achieved at each of the lower administrative levels (sub-counties, parishes, and villages) by sorting the sampling frame within each sampling stratum before sample selection, according to administrative units in different levels, and by using a probability proportional-to-size selection at the first stage of sampling. In the first stage, 697 Enumeration Areas (EA) were selected, 162 EA in urban and 535 in rural areas ""Fig 2,"". One cluster was eliminated due to disputed boundaries leaving a total of 696 clusters. The EAs were selected with probability proportional to the EA size and with independent selection in each sampling stratum. The EA size is the number of residential households residing in the EA based on the 2014 Uganda Population and Housing Census. Some of the selected EAs were large, with more than 250 households. To minimize the task of household listing, these large EAs were segmented, and only one segment, with probability proportional to the segment size, was selected for the survey. Household listing was conducted only in the selected segment. So, a 2016 UDHS cluster was either an EA or a segment of an EA. In the second stage of selection, a fixed number of 30 households per cluster were selected with an equal probability of systematic selection from the newly created household listing. To minimize bias, no replacements and no changes of the preselected households were allowed in the implementing stages. In total, a representative sample of 20,880 households was randomly

**Table A.3  Sample allocation of clusters and households by region and type of residence**

| Region | Number of clusters allocated | | | Number of households allocated | | |
|---|---|---|---|---|---|---|
| | Urban | Rural | Total | Urban | Rural | Total |
| Kampala | 45 | 0 | 45 | 1,350 | 0 | 1,350 |
| South Central - not island | 20 | 36 | 56 | 600 | 1,080 | 1,680 |
| North Central - not island | 12 | 33 | 45 | 360 | 990 | 1,350 |
| South Central - island | 2 | 10 | 12 | 60 | 300 | 360 |
| North Central - island | 2 | 12 | 14 | 60 | 360 | 420 |
| Busoga - not island | 7 | 31 | 38 | 210 | 930 | 1,140 |
| Busoga - island | 0 | 21 | 21 | 0 | 630 | 630 |
| Bukedi | 6 | 35 | 41 | 180 | 1,050 | 1,230 |
| Bugisu | 7 | 34 | 41 | 210 | 1,020 | 1,230 |
| Teso | 4 | 36 | 40 | 120 | 1,080 | 1,200 |
| Karamoja | 4 | 30 | 34 | 120 | 900 | 1,020 |
| Lango | 5 | 39 | 44 | 150 | 1,170 | 1,320 |
| Acholi | 8 | 32 | 40 | 240 | 960 | 1,200 |
| West Nile | 5 | 40 | 45 | 150 | 1,200 | 1,350 |
| Bunyoro | 8 | 36 | 44 | 240 | 1,080 | 1,320 |
| Tooro | 9 | 39 | 48 | 270 | 1,170 | 1,440 |
| Ankole | 12 | 37 | 49 | 360 | 1,110 | 1,470 |
| Kigezi | 6 | 34 | 40 | 180 | 1,020 | 1,200 |
| **Uganda** | **162** | **535** | **697** | **4,860** | **16,050** | **20,910** |

**Fig 2.  Allocation of sample clusters and households.**

selected for the UDHS 2016. All women aged 15–49 who were either permanent residents of the selected households or visitors who stayed in the household the night before the survey were interviewed.

## Data collection procedure

A structured and pre-tested questionnaire was used as a tool for data collection. The questionnaire was developed in English and then translated into nine different local languages. The questionnaire was developed based on standard DHS survey questionnaires and programmed into tablet computers to facilitate computer-assisted personal interviewing (CAPI) for data collection purposes, with the capability to choose any of the nine languages for each questionnaire. The UDHS and ICF technical teams trained 45 participants who administered the paper and electronic questionnaires with tablet computers. All trainees had some experience with household surveys. The technical teams conducted debriefing sessions with the pre-test field staff and modifications to the questionnaires were made based on lessons learned from the exercise. A total of 173 fieldworkers (108 women and 65 men) were recruited and trained to serve as supervisors, CAPI managers, interviewers, health technicians, and reserve interviewers for the main fieldwork. The training course included instruction on interviewing techniques and field procedures, a detailed review of questionnaire content, instruction on administering the paper and electronic questionnaires, mock interviews between participants in the classroom, and practice interviews with actual respondents in areas outside the 2016 UDHS sample. A two-day field practice was organized to provide trainees with additional hands-on practice before the actual fieldwork.

A total of 84 participants were selected to serve as interviewers, 21 as health technicians, 21 as field data managers, and 21 as team leaders. The selection of team leaders and field data managers was based on experience in leading survey teams and performance during the pre-test and main training. Supervisory activities included assigning households and receiving completed interviews from interviewers, recognizing and dealing with error messages, receiving system updates and distributing updates to interviewers, resolving duplicated cases, closing clusters, and transferring interviews to the central office via a secure Internet file streaming system (IFSS). Data collection was conducted by 21 field teams, each consisting of one team leader, one field data manager, three female interviewers, one male interviewer, one health technician, and one driver. Electronic data files were transferred from each interviewer's tablet computer to the team supervisor's tablet computer every day. The field supervisors transferred data to the central data processing office via IFSS. Senior staff from the Makerere University School of Public Health, the Ministry of Health, and UBOS, and a survey technical specialist. The DHS Program coordinated and supervised fieldwork activities. Data collection took place from 20 June 2016 through 16 December 2016.

## Variables and measurements

**Dependent variable.** The outcome variable of our study is the use of current family planning methods (traditional, folkloric, and modern methods) among women of reproductive age.

**Independent variables.** For this study, we used the women's questionnaire which collected information from women of reproductive age, 15–49 years. The women's questionnaire consisted of 12 sections, however, we used variables for four sections; section 1 –respondent's background, section 3 –contraception, section 9 –fertility preferences, and section 10 –husband's background and women's work. The independent variables used were: age (15–19, 20–

24, 25–29, 30–34, 35–39, 40–44, 45–49), place of residence (urban, rural), level of education (no education, primary, secondary, higher education), literacy level (cannot read at all, read parts of the sentence, read whole sentence, no card required, visually impaired), current marital status (never married, married, living with partner, widowed, divorced, separated), religious affiliation (anglican, catholic, muslim, Pentecostal, other), currently working (no, yes)husband's education (no education, primary, secondary, higher education, don't know, not applicable), wealth index (lowest, second, middle, fourth, highest) regions of Uganda (kampala, south Buganda, north Buganda, busoga, bukedi, bugisu, teso, karamoja, lango, acholi, west nile, bunyoro, tooro, ankole, kigezi), decision maker on use of family planning (mainly respondent, mainly husband, joint decision, others, not applicable), last source of FP user (government/pharmacy, community delivery, NGO, private clinic, private pharmacy, shop/church/friend, others, not applicable), current use of FP methods (no, yes), current use of methods (no method, folkloric method, modern method), current method type (not using any method, pill, IUD, male condom, female sterilization, periodic abstinence, withdrawal method, implant/norplant, lactational amenorrhea, injections, others), pattern of FP use (currently using, used since last birth, used before last birth, never used), intention to use contraceptive (using modern methods, using traditional methods, non-user intends to use later, used before last birth, never used), knowledge of ovulatory cycle (during her period, after period ended, middle of the cycle, before period begins, at any time, others, don't know), pregnancy after birth (yes, no), knowledge of family planning methods (yes, no), heard about FP on the radio last few months (yes, no) heard about FP on the television last few months (yes, no), heard about FP in newspaper last few months (yes, no), heard about FP by phone via text messages (yes, no), visited by field worker last 12 months (yes, no), visited health facility last 12 months (yes, no), told about FP at the health facility (yes, no).

## Data management and analysis

The downloaded data was entered into the SPSS software version 32.0 and data was cleaned, transformed to populate cells with few values, and re-coded as well. The collinearity effect was checked during bivariate analysis using a cut-off value of variance inflation factor (VIF) equal to and less than 4. The univariable, bivariable, and multivariable analyses were performed. The Univariable analysis was used to summarise the socio-demographic factors to find the pattern within the dataset meanwhile, the bivariable analysis was used to compare two variables to measure the relationship between them, and also identify variables to include in the regression analysis. On the other hand, multivariable analysis, which is a more complex analysis technique was used to understand the relationship between two or more variables and also control for confounding factors. At the univariable level, frequencies and proportions were determined. At the bivariable level, analysis was done by cross-tabulation using the Pearson Chi-Square ($x^2$) test for categorical variables. Pearson's Chi-square test was used because it is appropriate to analyze data with a binary outcome and independent categorical variables.

The associations between the outcome and independent variables were measured using the odds ratio (OR) for which a 95% confidence interval (CI) was calculated. All variables that showed a significant association of $p<0.05$ at the bivariable level were further analyzed at the multivariable level using a binary logistic regression. Binary logistic regression analysis was used because the data set is normally distributed and has a binary outcome. The adjusted odds ratio (aOR) was undertaken using a simultaneous modeling technique to determine the presence of associations between the outcome and independent variables. Model fitness was performed using the Hosmer and Lemeshow Chi-Square test at $p>0.05$.

## Results of the study

### Description of socio-demographic characteristics

A total of 18,506 samples of women of reproductive age (15–49 years) were included in the dataset where 23.1% were adolescents (15–19 years) and 76.3% lived in rural areas (Table 1). More than half (58.9%) attained a primary level of education, 31.4% were married, and 40.8% were affiliated with the Catholic religion. Further analysis revealed that the majority of women (73.9%) were currently working at the time of the interview and most of them (21.8%) lived in households with the highest wealth index. The Baganda tribe represented the highest proportion of ethnic groupings (13.2%) and only a handful of women (15.5%) used government clinics/pharmacies as the main source for family planning methods.

### Prevalence of family planning uptake

The prevalence of current FP use among women was 29.3%, with 26.6% of them using modern contraceptive methods (Table 2). Most women preferred injections (13.5%), followed by implants (4.8%) and male condoms (2.9%). Analysis of the pattern of FP use showed that less than half (29.3%) were currently using at least one method, 12.1% used since their last birth, 14.1% used before their last birth, and 44.3% never used any FP method.

### Determinants of family planning uptake

The table of adjusted analyses is indicated in Table 3. The adjusted analysis revealed significantly higher odds of current FP use among; women who were knowledgeable about their ovulatory cycle (aOR = 2.58, 95% CI: 1.07–6.26); Older women of age group 40–44 years (aOR = 2.09, 95% CI: 1.40–3.12); women who had attained a secondary level of education (aOR = 1.91, 95% CI: 1.32–2.76); those who lived in households with the highest wealth index (aOR = 1.87, 95% CI: 1.29–2.72); women who were aware of the availability of FP methods (aOR = 1.87, 95% CI: 1.04–3.37); and women living in the Lango sub-region of Uganda (aOR = 1.67, 95% CI: 1.05–2.67). The overall model shows a good fit of data with the Pearson Chi-Square test of p = 0.19 and the model is said to fit well when the p-value is more than 0.05.

## Discussion

### Discussions of key results

Uganda has the lowest FP prevalence rate as compared to the rates in neighboring countries like Kenya (45.5%), Rwanda (51.6%), and Tanzania (34.4%) [26]. The difference in the FP prevalence rates could be due to the low level of education among women, having three or more children, living in rural areas, husband's disagreement on contraceptive use, perceived side effects, infant mortality; negative traditional practices, knowledge gaps on contraceptive methods, fears, rumors, and misconceptions about specific methods and unavailability and poor quality of services [27]. A recent study on contraceptive use further alluded to cultural beliefs, financial constraints to access contraceptives, and limited sources of family planning information like television and newspapers [23]. The low prevalence of family planning use could negatively affect Uganda's progress in achieving sustainable development goal (SDG) 3 target 3.7 aimed at ensuring universal access to sexual and reproductive healthcare services, including family planning by 2030 if immediate interventions are not put in place [28].

Our study found significantly higher odds of FP use among older women in the age group 40–44 years. The previous analysis of the 2011 UDHS and Uganda FP costed implementation plan 2015–2020 revealed disparities in the use of family planning by; age, marital status, education,

**Table 1. Socio-demographic characteristics of women of reproductive age, UDHS 2016 (N = 18,506).**

| Respondent's characteristic | | Frequency (%) | Currently using FP methods (%) |
|---|---|---|---|
| Age group (years) | | | |
| | 15–19 | 4,276 (23.1) | 9.5 |
| | 20–24 | 3,782 (20.4) | 30.5 |
| | 25–29 | 3,014 (16.3) | 39.1 |
| | 30–34 | 2,600 (14.0) | 39.0 |
| | 35–39 | 2,029 (11.0) | 38.4 |
| | 40–44 | 1,621 (8.8) | 38.0 |
| | 45–49 | 1,184 (6.4) | 22.7 |
| Place of residence | | | |
| | Urban | 4,379 (23.7) | 33.0 |
| | Rural | 14,127 (76.3) | 28.1 |
| Highest level of education | | | |
| | No education | 2,071 (11.2) | 39.2 |
| | Primary | 10,893 (58.9) | 31.9 |
| | Secondary | 4,213 (22.8) | 28.7 |
| | Higher | 1,329 (7.2) | 20.6 |
| Current marital status | | | |
| | Never married | 4,738 (25.6) | 11.5 |
| | Married | 5,813 (31.4) | 39.4 |
| | Living with partner | 5,566 (30.1) | 35.5 |
| | Widowed | 523 (2.8) | 15.1 |
| | Divorced | 139 (0.8) | 19.4 |
| | Separated | 1,727 (9.3) | 29.0 |
| Religious affiliation | | | |
| | Anglican | 5,799 (31.3) | 31.6 |
| | Catholic | 7,552 (40.8) | 28.0 |
| | Muslim | 2,166 (11.7) | 30.4 |
| | Pentecostal | 2,436 (13.2) | 28.2 |
| | Other[a] | 553 (3.0) | 27.0 |
| Currently working | | | |
| | No | 4,838 (26.1) | 32.6 |
| | Yes | 13,668 (73.9) | 19.9 |
| Husband's educational level | | | |
| | No education | 882 (4.8) | 47.5 |
| | Primary | 6,006 (32.5) | 40.7 |
| | Secondary | 2,909 (15.7) | 36.3 |
| | Higher | 1,287 (7.0) | 21.5 |
| Wealth index | | | |
| | Lowest | 3,884 (21.0) | 34.7 |
| | Second | 3,640 (19.7) | 34.6 |
| | Middle | 3,485 (18.8) | 30.6 |
| | Fourth | 3,454 (18.7) | 27.3 |
| | Highest | 4,043 (21.8) | 19.6 |
| Region in Uganda | | | |
| | Kampala | 1,300 (7.0) | 31.9 |
| | South Buganda | 1,615 (8.7) | 35.4 |
| | North Buganda | 1,410 (7.6) | 35.6 |

*(Continued)*

**Table 1.** (Continued)

| Respondent's characteristic | | Frequency (%) | Currently using FP methods (%) |
|---|---|---|---|
| | Busoga | 1,530 (8.3) | 26.3 |
| | Bukedi | 1,205 (6.5) | 32.9 |
| | Bugisu | 957 (5.2) | 36.1 |
| | Teso | 1,347 (7.3) | 25.6 |
| | Karamoja | 741 (4.0) | 6.6 |
| | Lango | 1,236 (6.7) | 32.8 |
| | Acholi | 1,110 (6.0) | 23.9 |
| | West Nile | 1,281 (6.9) | 15.8 |
| | Bunyoro | 1,213 (6.6) | 26.5 |
| | Tooro | 1,301 (7.0) | 35.3 |
| | Ankole | 1,301 (7.0) | 32.7 |
| | Kigezi | 959 (5.2) | 32.8 |
| Decision makers on the use of family planning | | | |
| | Mainly respondent | 1,294 (7.0) | - |
| | Mainly husband/partner | 311 (1.7) | - |
| | Joint decision | 2,655 (14.3) | - |
| | Other[b] | 8 (0.0) | - |
| | Not applicable | 14,238 (76.9) | - |
| Last source of family planning user | | | |
| | Government/pharmacy | 2,869 (15.5) | - |
| | Community delivery | 222 (1.2) | - |
| | NGO | 70 (0.4) | - |
| | Private clinic | 1,500 (8.1) | - |
| | Private pharmacy | 319 (1.7) | - |
| | Shop, church, friend | 170 (0.9) | - |
| | Others | 268 (1.4) | - |
| | Not applicable | 13,088 (70.7) | - |

UDHS = Uganda demographic and health survey; N = the total number of respondents; fp = family planning;
NGO = non-governmental organization
other[a] = Seventh Day Adventist, orthodox, Bahai, Baptist, Jewish, Presbyterian, Mammon, Hindu, Buddhist,
Jehovah's witness, Salvation army, & Traditional; and
other[b] = family planning clinic, mobile clinic, community health worker, private sector, and private doctor.

socio-economic status, and rural-urban geographic location [29]. Our finding concurs with other studies that found the use of FP increases with older age [30, 31]. It is believed that older women are more exposed to information concerning childbearing and the dangers of high parities so they have appreciated the importance of the uptake of family planning methods.

Educational achievements of both women and their husbands were found in some studies to be very significant factors in the use of FP methods [31, 32]. However, the present study found the education of women alone as the driver of the FP uptake. Unlike women with no formal education, women with at least secondary education are more likely users of FP methods [33]. This does not come as much of a surprise as higher education attainment increases female decision-making powers and awareness of the benefits of good family planning practices. This affirms the relevance of education in matters concerning the use of FP in Uganda. Thus, the dispute in Uganda that universal secondary education (USE) works towards enhancing FP use is highly supported.

**Table 2. Current family planning uptake among women of reproductive age, UDHS 2016.**

| Respondent characteristics | | Frequency (n) | Percent (%) |
|---|---|---|---|
| Current use of FP methods | | | |
| | No | 13,088 | 70.7 |
| | Yes | 5,418 | 29.3 |
| Current use by methods | | | |
| | No method | 13,088 | 70.7 |
| | Folkloric method | 46 | 0.2 |
| | Traditional method | 458 | 2.5 |
| | Modern method | 4,914 | 26.6 |
| Current method type | | | |
| | Not using any method | 13,088 | 70.7 |
| | Pill | 256 | 1.4 |
| | IUD | 197 | 1.1 |
| | Male condom | 534 | 2.9 |
| | Female sterilization | 355 | 1.9 |
| | Periodic abstinence | 169 | 0.9 |
| | Withdrawal | 289 | 1.6 |
| | Implant/Norplant | 858 | 4.8 |
| | Lactational amenorrhea | 108 | 0.6 |
| | Injections | 2,506 | 13.5 |
| | Other | 116 | 0.6 |
| Pattern of family planning use | | | |
| | Currently using | 5,418 | 29.3 |
| | Used since last birth | 2,247 | 12.1 |
| | Used before last birth | 2,635 | 14.1 |
| | Never used | 8,206 | 44.3 |
| Intention to use contraceptive | | | |
| | Using modern method | 4,914 | 26.6 |
| | Using traditional method | 504 | 2.7 |
| | Non-user intends to use later | 7,879 | 42.6 |
| | Does not intend to use | 5,209 | 28.1 |

n = frequency and % = percent

In our study, women living in households with the highest wealth index had significantly higher odds of FP. This finding concurs with a study conducted in Ethiopia [34] but, contradicts another study conducted in Uganda which found that wealth was not associated with FP use [35]. This variation can be explained by current women's empowerment through education and media awareness. The poor are less likely to be well informed about FP methods which can be attributed to a lack of ownership of television sets, mobile phones, or buying newspapers which limits getting FP information [36–38]. The poor may also have problems accessing healthcare due to long distances to the health facility, lack of money for transportation, and limited access to FP as a result of out-of-pocket expenditures to purchase FP methods [39].

Our finding found geographical differences to be associated with the FP uptake among women of reproductive age. Specifically, women in the Lango sub-region were more likely users of FP methods as compared to other regions of Uganda. This finding is in line with other studies that have also shown geographical differences to influence FP uptake [39–41]. This is possible considering some of the interventions by the government and other development

**Table 3. Adjusted analysis of current family planning use among women of reproductive age.**

| Respondent's characteristic | | Current use of FP methods | | Adjusted ORs |
|---|---|---|---|---|
| | | Yes (%) | No (%) | (95% CI) |
| Age group (years) | | | | |
| | 15–19 | 405 (9.5) | 3,871 (90.5) | 1.01 (0.62–1.64) |
| | 20–24 | 1,155 (30.5) | 2,627 (69.5) | 1.28 (0.89–1.87) |
| | 25–29 | 1,178 (39.1) | 1,836 (60.9) | 1.52* (1.05–2.21) |
| | 30–34 | 1,015 (39.0) | 1,585 (61.0) | 1.61** (1.11–2.33) |
| | 35–39 | 780 (38.4) | 1,249 (61.6) | 1.81** (1.23–2.66) |
| | 40–44 | 616 (38.0) | 1,005 (62.0) | 2.09*** (1.40–3.12) |
| | 45–49 | 269 (22.7) | 915 (77.3) | 1 |
| Place of residence | | | | |
| | Urban | 1,445 (33.0) | 2,934 (67.0) | 1.18 (0.91–1.53) |
| | Rural | 3,973 (28.1) | 10,154 (71.9) | 1 |
| Highest level of education | | | | |
| | Higher | 521 (39.2) | 808 (60.8) | 1.70* (1.03–2.81) |
| | Secondary | 1,344 (31.9) | 2,869 (68.1) | 1.91*** (1.32–2.76) |
| | Primary | 3,127 (28.7) | 7,766 (71.3) | 1.73*** (1.26–2.36) |
| | No education | 426 (20.6) | 1,645 (79.4) | 1 |
| Current marital status | | | | |
| | Never married | 543 (11.5) | 4,195 (88.5) | - |
| | Married | 2,291 (39.4) | 3,522 (60.6) | - |
| | Living with partner | 1,977 (35.5) | 3,589 (64.5) | - |
| | Widowed | 79 (15.1) | 444 (84.9) | - |
| | Divorced | 27 (19.4) | 112 (80.6) | - |
| | Separated | 501 (29.0) | 1,226 (71.0) | - |
| Religious affiliation | | | | |
| | Anglican | 1,833 (31.6) | 3,966 (68.4) | 1.19 (0.90–1.56) |
| | Catholic | 2,113 (28.0) | 5,439 (72.0) | 1.03 (0.78–1.35) |
| | Muslim | 658 (30.4) | 1,508 (69.6) | 0.91 (0.65–1.28) |
| | Other | 156 (28.2) | 397 (71.8) | 0.99 (0.57–1.73) |
| | Pentecostal | 658 (27.0) | 1,778 (73.0) | 1 |
| Currently working | | | | |
| | Yes | 4,455 (32.6) | 9,213 (67.4) | 1.25 (0.99–1.59) |
| | No | 963 (19.9) | 3,875 (80.1) | 1 |
| Husband's educational level | | | | |
| | Higher | 611 (47.5) | 676 (52.5) | 1.54 (0.95–2.49) |
| | Secondary | 1,184 (40.7) | 1,725 (59.3) | 1.17 (0.77–1.79) |
| | Primary | 2,181 (36.3) | 3,825 (63.7) | 1.13 (0.76–1.68) |
| | No education | 190 (21.5) | 692 (78.5) | 1 |
| Wealth index | | | | |
| | Highest | 1,401 (34.7) | 2,642 (65.3) | 1.87*** (1.29–2.72) |
| | Fourth | 1,196 (34.6) | 2,258 (65.4) | 1.77*** (1.47–2.64) |
| | Middle | 1,066 (30.6) | 2,419 (69.4) | 1.73*** (1.31–2.27) |
| | Second | 993 (27.3) | 2,647 (72.7) | 1.31* (1.01–1.69) |
| | Lowest | 762 (19.6) | 3,122 (80.4) | 1 |
| Regions in Uganda | | | | |
| | Kampala | 415 (31.9) | 885 (68.1) | 0.69 (0.35–1.36) |
| | South Central | 571 (35.4) | 1,044 (64.6) | 0.93 (0.56–1.54) |

(*Continued*)

**Table 3.** (Continued)

| Respondent's characteristic | | Current use of FP methods | | Adjusted ORs |
|---|---|---|---|---|
| | | Yes (%) | No (%) | (95% CI) |
| | North Central | 502 (35.6) | 908 (64.4) | 1.04 (0.61–1.79) |
| | Busoga | 403 (26.3) | 1,127 (73.7) | 0.63 (0.41–0.98) |
| | Bukedi | 396 (32.9) | 809 (67.1) | 1.03 (0.67–1.57) |
| | Bugisu | 345 (36.1) | 612 (63.9) | 0.99 (0.60–1.64) |
| | Teso | 345 (25.6) | 1,002 (74.4) | 1.16 (0.74–1.82) |
| | Karamoja | 49 (6.6) | 692 (93.4) | 0.23***(0.12–0.42) |
| | Lango | 405 (32.8) | 831 (67.2) | 1.67* (1.05–2.67) |
| | Acholi | 265 (23.9) | 845 (76.1) | 0.53* (0.30–0.96) |
| | West Nile | 202 (15.8) | 1,079 (84.2) | 0.51**(0.33–0.80) |
| | Bunyoro | 321 (26.5) | 892 (73.5) | 0.88 (0.54–1.45) |
| | Tooro | 459 (35.3) | 842 (64.7) | 1.03 (0.63–1.67) |
| | Ankole | 425 (32.7) | 876 (67.3) | 1.01 (0.65–1.57) |
| | Kigezi | 315 (32.8) | 644 (67.2) | 1 |
| Literacy level | | | | |
| | Cannot read at all | 1,653 (26.1) | 4,671 (73.9) | - |
| | Read parts of sentence | 635 (28.3) | 1,607 (71.7) | - |
| | Read whole sentence | 3,102 (31.6) | 6,708 (68.4) | - |
| | No card required | 24 (24.5) | 74 (75.5) | - |
| | Visually impaired | 4 (12.5) | 28 (87.5) | - |
| Knowledge of ovulatory cycle | | | | |
| | During her period | 46 (20.1) | 183 (79.9) | 2.58*(1.07–6.26) |
| | After period ended | 2,595 (31.4) | 5,674 (68.6) | 1.37 (0.95–1.98) |
| | Middle of the cycle | 1,377 (34.3) | 2,641 (65.7) | 1.34 (0.92–1.97) |
| | Before period begins | 596 (32.1) | 1,259 (67.9) | 1.41 (0.91–2.17) |
| | At any time | 393 (22.1) | 1,386 (77.9) | 1.25 (0.77–2.03) |
| | Other* | 44 (34.4) | 84 (65.6) | 1.24 (0.46–3.35) |
| | Don't know | 367 (16.5) | 1,861 (83.5) | 1 |
| Can a woman be pregnant after birth? | | | | |
| | Yes | 3,524 (35.8) | 6,311 (64.2) | 1.77* (1.10–2.85) |
| | No | 1,634 (24.5) | 5,043 (75.5) | 1.38 (0.85–2.24) |
| | Don't know | 260 (13.0) | 1,734 (87.0) | 1 |
| Knowledge of family planning methods | | | | |
| | Yes | 5,342 (31.1) | 11,835 (68.9) | 1.87* (1.04–3.37) |
| | No | 76 (5.7) | 1,253 (94.3) | 1 |
| Heard of FP on the radio in the last few months | | | | |
| | Yes | 3,780 (31.8) | 8,113 (68.2) | 1.41***(1.17–1.72) |
| | No | 1,638 (24.8) | 4,975 (75.2) | 1 |
| Heard of FP on television in the last few months | | | | |
| | Yes | 1,160 (35.4) | 2,115 (64.6) | 0.96 (0.72–1.28) |
| | No | 4,258 (28.0) | 10,973 (72.0) | 1 |
| Heard of FP in the newspaper in the last few months | | | | |
| | Yes | 645 (33.1) | 1,302 (66.9) | 0.73 (0.53–1.01) |
| | No | 4,773 (28.8) | 11,786 (71.2) | 1 |
| Heard of FP by phone via text messages | | | | |
| | Yes | 201 (39.5) | 308 (60.5) | 1.20 (0.80–1.77) |
| | No | 5,217 (29.0) | 12,780 (71.0) | 1 |

(*Continued*)

**Table 3.** (Continued)

| Respondent's characteristic | | Current use of FP methods | | Adjusted ORs |
|---|---|---|---|---|
| | | Yes (%) | No (%) | (95% CI) |
| Visited by a field worker in the last 12 months | | | | |
| | Yes | 1,548 (31.0) | 3,450 (69.0) | 0.98 (0.82–1.18) |
| | No | 3,870 (28.6) | 9,638 (71.4) | 1 |
| Filed worker talked about other FP methods | | | | |
| | Yes | 518 (31.2) | 1,142 (68.8) | 1.05 (0.89–1.25) |
| | No | 1,030 (30.9) | 2,308 (69.1) | 1 |
| Visited health facility in the last 12 months | | | | |
| | Yes | 3,999 (31.4) | 8,753 (68.6) | - |
| | No | 1,419 (24.7) | 4,335 (75.3) | - |
| Told about FP at the health facility | | | | |
| | Yes | 1,724 (33.4) | 3,437 (66.6) | - |
| | No | 2,275 (30.0) | 5,316 (70.0) | - |
| Model fitness test ($x^2$) | | | | P = 0.19 |

***P<0.001

**P<0.01

*P<0.05, ORs = odds ratios, CI = confidence interval, FP = family planning, other* = wasn't provided in the notes.

agencies operating in the region that have yielded a positive impact on enhancing access to and use of FP methods. For example, the use of the voucher-plus system ensured that poor women had access to quality maternal health care and FP services at a reduced cost. Furthermore, some regions systematically fail to benefit from wider improvements in health experienced by the general population as such groupings are geographically or linguistically remote or benefit selectively from national and international investments [42].

The study showed knowledge as a strong predictor of current family planning use. This was also corroborated by Olugbenga et al (2011) in their study carried out in South-Western Nigeria [43]. They noted that this pattern should be expected considering much enlightenment that is ongoing on the use of FP in the country. Education exposes women to reproductive health information and empowers them to make appropriate judgments. It is however worth noting that some family planning methods were unpopular among respondents because they were not readily available and relatively more expensive than other methods. These included male sterilization (vasectomy), female sterilization (tubal ligation), lactational amenorrhea, intra-uterine device (IUD) levonorgestrel, and vaginal rings.

This study observed that awareness through listening to the local radio was the predominant source of information for FP methods among women. This finding is in tandem with findings elsewhere that have documented the importance of information in influencing FP [26]. The use of public media sources like listening to the radio, watching television, and reading newspapers increases the awareness of people on FP methods. This present study did not find any significant association between FP uptake and watching television, reading newspapers, and receiving text messages via mobile phones. This may be so because the majority of women are illiterate and live in rural areas where they have access to local radios but not television, newspapers, and mobile phones. However, to improve FP use, media in all its forms play a major role in influencing the usage of FP methods [44]. Information gives women the freedom of choice and can enable them to make better choices of FP methods in addition to having an opportunity to discuss with their spouses.

### Implications of findings

Increased uptake of FP methods brings significant health and other benefits. It further offers a range of potential non-health benefits that encompass expanded education opportunities and empowerment for women, and sustainable population growth and economic development of Uganda.

### Strengths and limitations of the study

A major strength of this study is that the data used are nationally representative of women of reproductive age in the entire country, Uganda. A major limitation is that the data are cross-sectional and the authors could not establish temporality between participants' exposure to some of the independent variables (i.e., program exposure) and the outcome. Since it was a population-based study, health facility factors influencing the use of FP methods were not included in the survey.

## Conclusions

Our study shows the prevalence of current FP use in Uganda to be low and a threat to achieving the SDG 3 target of 3.7.1 by 2030. In conclusion, the study suggests that improvement in women's education attainment, socio-economic position, and awareness may help increase contraceptive use in the population. Policymakers need to amend the existing policies to promote the use of FP methods among young women since they are fewer users of FP methods as compared to older women.

## Acknowledgments

We appreciate the USAID DHS Programme for granting us the use of the dataset.

## Author Contributions

**Conceptualization:** Anthony Mark Ochen.

**Formal analysis:** Anthony Mark Ochen.

**Methodology:** Anthony Mark Ochen, Che Chi Primus.

**Visualization:** Che Chi Primus.

**Writing – original draft:** Anthony Mark Ochen, Che Chi Primus.

**Writing – review & editing:** Anthony Mark Ochen, Che Chi Primus.

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
