## [Decision Letter · Decision Letter 0]

16 Feb 2023

PGPH-D-22-01403

Family planning uptake and its associated factors among women of reproductive age in Uganda: an insight from the Uganda Demographic and Health Survey 2016

Dear Dr. Ochen,

Thank you for submitting your manuscript to PLOS Global Public Health. After careful consideration, we feel that it has merit but does not fully meet PLOS Global Public Health’s publication criteria as it currently stands. Therefore, we invite you to submit a revised version of the manuscript that addresses the points raised during the review process.

EDITOR:

In line with publication criteria it is important that the authors:

- Address the methodological issues raised

- Ensure that conclusions are in line with the presented data

We look forward to receiving your revised manuscript.

Kind regards,

Claire E. von Mollendorf

Academic Editor

Journal Requirements:

1. Please provide separate figure files in .tif or .eps format only and remove any figures embedded in your manuscript file. Please also ensure that all files are under our size limit of 10MB.

2. We have noticed that you have a list of Supporting Information legends in your manuscript. However, there are no corresponding files uploaded to the submission. Please upload them as separate files with the item type 'Supporting Information'. 

Additional Editor Comments (if provided):

Reviewers' comments:

Reviewer's Responses to Questions

**Comments to the Author**

1. Does this manuscript meet PLOS Global Public Health’s publication criteria? Is the manuscript technically sound, and do the data support the conclusions? The manuscript must describe methodologically and ethically rigorous research with conclusions that are appropriately drawn based on the data presented.

Reviewer #1: Yes

Reviewer #2: Partly

2. Has the statistical analysis been performed appropriately and rigorously?

Reviewer #1: No

Reviewer #2: No

3. Have the authors made all data underlying the findings in their manuscript fully available (please refer to the Data Availability Statement at the start of the manuscript PDF file)?

Reviewer #1: Yes

Reviewer #2: Yes

4. Is the manuscript presented in an intelligible fashion and written in standard English?

Reviewer #1: Yes

Reviewer #2: Yes

5. Review Comments to the Author

Reviewer #1: Introduction

• Need to include socio-economical parameters in Uganda to be able to align with the findings. Example proportion of women with formal education, poverty rate, annual expenditure on health.

• Need to include unmet need for women of reproductive age in Uganda

• Need to explain current situation of the known factors associated with FP uptake.

• Need to expand on the explanation of independent variables selected in this study.

• Rationale for the study need to revised

Methods

Independent variables:

• Need to define and explain how wealth index was calculated, decision making, knowledgeable about ovulatory cycle

• Advise to consider gender related factors that could explain uptake of FP in Uganda. Example male engagement (accompany to FP clinics, Couple discussion on FP use). Cultural aspects like seeking prior approval from husband to use family planning. These aspects of information are usually collected in DHS

Data analysis

• Since the sampling procedure was multi-stage stratified cluster sampling, the authors should clearly show how did they took into account on the effect of intra and inter cluster variations.

Results

Result of variable ‘’ Decision maker for using contraceptive’’ was not shown despite being mentioned in the methods section under independent variables

Reviewer #2: Abstract

First sentence: pls mention when FP use was “still low at 30%” (i.e., the year).

Is the DHS sample limited to married women of repro age or all women of repro age?

When you say “awareness of the availability of family planning methods” do you mean awareness of where and how to access family planning or awareness of different methods?

Since you identified cross-sectional associations, I think you should consider toning down the implications. You conclusion sentence could say, instead, that “the study suggests that improvement in women’s education attainment, socio-economic position and awareness may help increase increases in contraceptive use in the population.”

Be consistent with use of “family planning” versus “FP”.

Introduction

Authors should number the lines on each page to facilitate the review.

In the first para, to strengthen your argument you may wish to reference a study that links preventable maternal mortality with the prevalence of unintended pregnancies, which underscores importance of family planning as a maternal health intervention.

Second para: attainment of desired family size can be achieved through natural family planning practices as well as the use of “FP methods” (which, I understood here to mean modern contraception).

Of the 1.9 billion WRA, did 1.1 billion report that they needed FP methods or that they had a desire to space or cease childbearing?

Final paragraph of page two/first para of page 3, where the explanations for the slow increase in global CPR. The authors should consider evoking some other arguments, e.g., sustained preferences for large family size. Or clarify that these explanations are supply side factors that tend to undermine the strength of FP programs.

Page 3 where it is written: “Therefore, universal access to FP services is an important global strategy to control fast-growing populations and improve maternal and child health.” I think the authors should be careful regarding the language that they use. Underscoring the importance of FP for “population control” has negative connotations (implies that reproductive coercion may be acceptable). They should emphasize that FP use should remain voluntary, but that access to a wide range of contraceptive methods for women’s choosing may enhance their health prospects and have wider benefits for their societies’ social and economic development.

“Furthermore, in 2019, over 190 million (10%) of married women were estimated to have an unmet need for FP where the prevalence was higher in Africa compared to other parts of the world (9).”

- Prevalence of what? Contraceptive use or unmet need?

“This is well articulated in the sustainable development goals (SDG) 3, target 3.7 calls on countries “by 2030, to ensure universal access to sexual and reproductive health-care services, including for FP, information and education, and the integration of reproductive health into national strategies and program”; with specifically 3.7.1 which calls for universal access to FP services to ensure healthy lives and well-being (10).”

- Should there be an ‘s’ at the end of program?

“Despite the government efforts to reduce high fertility levels and increase uptake of FP services in Uganda, the prevalence rate is only 30% among married women which is the lowest in the East African region.”

- Please provide reference for this statement.

“Factors contributing to the low FP rates are multi-factorial and includes…”

- Should be “and include…”

Page 3-4: “Monitoring factors influencing the uptake of FP services is important to target scarce public resources to those with more need and enhance the progress towards achieving the global targets.”

- This sentence is confusing and should be re-written.

The review of the literature on this subject in Uganda is a bit thin. Only one study is referenced. The authors should situate their study in a more extensive review of the available literature. It does not have to be terribly long, but something more than just mentioning one study.

Materials and methods.

Provide a reference for the DHS.

Page 5: “To generate statistics that were representative of the country as a whole in the 15 regions, the number of women surveyed in each region contributed to the size of the total sample in proportion to the size of the region.”

- I think you mean that “to generate a sample that was representation of the country.

- In proportion to the size (geographic?) of the region or the population size of each region?

In the study design section, I would mention that the data collection method were household surveys and women were interviewed at home.

What is meant by this sentence: “Only one segment was selected for the survey with probability proportional to size, and the household listing was conducted only in the selected segment.”

A question I have is whether the total number of EA per region were selected on a population proportion to size basis, or whether within all EA the numbers of households selected per region were selected on a PPS basis?

I am a bit confused as to which aspects of the sampling strategy involved proportional allocation of sampling units to each region and which aspects were non-proportional. The first para of page 5 implies that the number of women selected per region was proportional to that regions overall contribution to the total size of the WRA population; however, here you write (on in the last para of page 6), “Due to the non-proportional allocation of the sample to different regions and their urban and rural areas, and the possible differences in response rates, sampling weights were analysed using the UDHS 2016 data to ensure that the survey results were representative at the national level as well as at the regional and sub-regional levels.”

I think that a clearer description of the sampling procedures is needed and this can reference the UDHS 2016 report where I think this is explained more clearly.

Page 8: “The women’s questionnaire consisted of 12 sections, however, we used variables for four sections; section 1 – respondent’s background, section 3 – contraception, section 9 – fertility preferences, and section 10 – husband’s background and woman’s work”

- I think you mean women’s work.

Data management and analysis- provide a reference that justification the VIF cut-off value of 6. This seems high to me. And I would explain procedurally how you applied the VIF cut off factor (e.g., you performed the procedure and identified all the covariates with a VIF >=6 and removed those from the model. Did you look at VIF values before or after conducting the bivariate tests of association (those estimate VIF for only the selected covariates as opposed to all of them)?

I would explain how the independent variables were measured (i.e., is level of education a continuous measure of number of years of schooling obtained or a categorical variable for no education, primary only, some secondary or more education). Similarly, for the wealth index, is one lowest or highest.

“All variables that showed a significant association of p < 0.05 at the bivariable level were further analysed at the multivariable level using a binary logistic regression. Binary logistic regression analysis was used because the data set is normally distributed and has a binary outcome. Some variables were also included in the multivariable model because these variables showed influence on the outcome variable, hence we needed to identify whether each of them had been confounded by another variable or not.”

- I found this statement confusing. It appears that you included some covariates in the multivariate model because they had a statistically significant bivariate association with the outcome, and that you included others because they “showed influence on the outcome variable” presumably this influence did not bear out in the bivariate tests but you included them anyway? I think the authors need explain more clearly how they selected covariates to include in the final model.

The earlier discussion on sample weights implies that the authors used a design-based modelling approach versus a model-based approach that would presumably address issues of clustering by incorporating random effects into the model. I think a more detailed discussion of how the sample weights were incorporated into the statistical modelling procedure is needed to demonstrate how the multi-level clustering effects were addressed in the analysis. To help with this, I think the authors should provide the equation for the final, multivariable model.

Results of the study

Description of socio-demographic characteristics of the sample is fine.

To simplify table 3 (which is too crowded and should only describe adjusted OR, in my opinion), I think that the breakdown of current FP use (yes/no) can be incorporated into Table 1 (Frequency and % can be combined in one columns with the frequency written and % included in parenthesis, and in an additional column you can put the % that are currently using an FP method).

I think that this section should describe some of the results of the bivariate tests, and include a table of those results as a supplementary file. The rationale for the authors’ decision to include the retained independent variables in the multivariate tests of adjusted associations with the outcome should be clear (a sentence that reports the variables that were ultimately retained), but we do not need to see every bivariate result in the main body of the paper.

I suspect that the models overfit. The decision to set the VIF cut off at 6 seemed high to me initially, and my review of the results reinforces this impression. In my experience (and after some cursory web searching I have done as I review the paper,) VIF <4 indicate low correlation among variables under ideal conditions, and VIF <5 is a maximum cutoff. The authors can dispute this but they need to justify the covariate selection more rigorously.

Some initial impressions from Table 3:

- Be careful about including covariates that are “on the causal pathway” between another independent variable and the outcome. For example, dual household incomes are likely to drive higher levels of socio-economic attainment, which, in turn, promotes conditions in which women may access FP. I would be cautious about including both variables (women currently working and household wealth index) in the same model. Doing so can be justified in some analysis that want to estimate the different in the magnitude of effect sizes produced by models that adjust and don’t adjust for mediating factors, but this does not seem to be the case here. So the authors should map out relationships between the independent variables and outcome and think through whether adjusting for mediating variables makes sense.

- The authors may be able to clarify this after they explain how they incorporated sample weights into the modelling procedure, but to me it is unusual to see region of residence included as a fixed effect if the sampling probabilities by region were used to develop the sample weights that were used in analysis. My intuition would be to drop the regional fixed effect.

- The inclusion of the final 10 or so covariates listed in table 3 raises some questions. Recall that this is a cross-sectional analysis. If the authors cannot demonstrate that participants’ exposure to interventions (e.g., communication interventions that disseminate FP information via radio, TV, newspapers) occurred before the occurrence of the outcome then they should be careful about using it in the model. The reason for this is reverse causality… a woman using FP might be more likely to tune into radio programs that discuss FP because they want to learn about more about the practice that they decided to uptake; a woman is more likely to get FP text messages if she is already a method user because the sender knows that this woman needs info on side effects or a reminder to come back for reinjection or resupply of other methods).

- A second reason to be concerned about included these 10 or so variables is selection bias… the selection mechanism that places women into levels of the exposure (e.g., exposure to the radio or TV program) may be very similar to the mechanisms that place them into levels of the outcome. In some instances, I think that the authors address this to some extent by adjusting for some socio-demographic characteristics (e.g., if a radio program targets youth because they are least likely to use FP, there is a common selection mechanisms for the exposure and outcome which is in part addressed by adjusting for age)… but this doesn’t address the challenge completely and introduces issues I mention about above adjusting for factors that are on the causal pathway (mediators). So, the authors should think carefully about the argument they are trying to make and decide whether including these covariates is really needed. I would encourage them to consider omitting these variables.

On page nine the authors state that “multivariable analysis, which is a more complex analysis technique was used to understand interactions between two or more variables and also control for confounding factors.”

- I did not see any discussion of interaction (i.e., effect modification) in the methods and results.

Discussion

I think that once the methodological issues I mention above are addressed, the analysis will be simpler and yield clearer messages to reflect upon in the discussion section. As it is, the discussion is solid, but the findings are very similar to what is already well known about the determinants of FP use in the sub-Saharan African region. So, readers are likely to ask at the end of the paper “so what?”

On page 3, the authors mention that it is important to monitor the determinants of FP prevalence to guide how scarce public resources are allocated. This is true and I think the discussion section should elaborate on how the findings can help decision-makers use resources to strengthen the Ugandan FP program. This is implied in the discussion, but a more detailed examination of the Ugandan FP landscape and program, and how specific findings illuminate areas for improvement, or can guide the implementation of programmatic improvements, would strengthen the paper. In other words, the discussion section should do more to convince readers that this update of the determinants of FP use in Uganda is practically useful at this time.

6. PLOS authors have the option to publish the peer review history of their article (what does this mean?). If published, this will include your full peer review and any attached files.

**Do you want your identity to be public for this peer review?** For information about this choice, including consent withdrawal, please see our Privacy Policy.

Reviewer #1: No

Reviewer #2: **Yes: **Colin Baynes

---

## [Decision Letter · Decision Letter 1]

20 Jun 2023

PGPH-D-22-01403R1

Family planning uptake and its associated factors among women of reproductive age in Uganda: an insight from the Uganda Demographic and Health Survey 2016

Dear Dr. Ochen,

Thank you for submitting your manuscript to PLOS Global Public Health. After careful consideration, we feel that it has merit but does not fully meet PLOS Global Public Health’s publication criteria as it currently stands. Therefore, we invite you to submit a revised version of the manuscript that addresses the points raised during the review process.

We look forward to receiving your revised manuscript.

Kind regards,

Claire E. von Mollendorf

Academic Editor

Journal Requirements:

1. We have noticed that you have a list of Supporting Information legends in your manuscript. However, there are no corresponding files uploaded to the submission. Please upload them as separate files with the item type 'Supporting Information'. 

Additional Editor Comments (if provided):

Reviewers' comments:

Reviewer's Responses to Questions

**Comments to the Author**

1. If the authors have adequately addressed your comments raised in a previous round of review and you feel that this manuscript is now acceptable for publication, you may indicate that here to bypass the “Comments to the Author” section, enter your conflict of interest statement in the “Confidential to Editor” section, and submit your "Accept" recommendation.

Reviewer #1: All comments have been addressed

Reviewer #2: (No Response)

2. Does this manuscript meet PLOS Global Public Health’s publication criteria? Is the manuscript technically sound, and do the data support the conclusions? The manuscript must describe methodologically and ethically rigorous research with conclusions that are appropriately drawn based on the data presented.

Reviewer #1: Yes

Reviewer #2: Partly

3. Has the statistical analysis been performed appropriately and rigorously?

Reviewer #1: Yes

Reviewer #2: No

4. Have the authors made all data underlying the findings in their manuscript fully available (please refer to the Data Availability Statement at the start of the manuscript PDF file)?

Reviewer #1: Yes

Reviewer #2: Yes

5. Is the manuscript presented in an intelligible fashion and written in standard English?

Reviewer #1: Yes

Reviewer #2: Yes

6. Review Comments to the Author

Reviewer #1: (No Response)

Reviewer #2: Commenting on the version of the paper with tracked changes.

Intro:

1. The first sentence of the intro does not make sense grammatically.

2. In the third sentence starting on line 39 or page, clarify what “it” is? I would provide a reference for this and possibly tone down description of the link between birth spacing and maternal death since most of the evidence on this is observation. But the point is well taken.

3. Line 47- “Uganda aspires to become a middle-income country by 2014”… we are now in 2023… so revise the sentence accordingly.

4. The assertions made between line 59 and 62 should be referenced.

5. Lines 63-68. Please clarify whether the literature you referenced state that 1.11 billion women expressed a need for contraceptive methods or a need to space/cease future pregnancy, and whether the 270 million had an unmet demand for contraception or an unmet desire to space/cease future pregnancy. The distinction is important because not all women that want to space/cease childbearing also want to use a contraceptive method.

6. Regarding the revision on lines 70-71 – I would give a reference that meager increase in global CPR is due to sustained preferences for large family size. My understanding is that this is really only true for sub-Saharan Africa. The factors that the authors cite as reasons for the “sustained preferences for large family size” seem more like reasons why women/families might not use contraception. Reasons to desire a large family, I would think, would have to do with the endurance of traditional cultural norms around family formation, the belief that offspring contribute more than they extract in terms of household wealth, etc.

7. Lines 78-82. I agree with the authors point, but it could be expressed more clearly, e.g., “For this reason, numerous scholars have pointed out that promoting voluntary access to a wide variety of contraceptive methods for women is an important component of countries’ strategies to advance social and economic development.”

8. I think the intro has all the needed details, but its structure is a bit confusing. It starts by pointing out the importance of family planning as a health intervention, then describes the profile of Uganda, and then circles back to explain why FP is important. I think the paper would benefit from a slight reorganization of the intro so that it describes (1) the health and demographic background of Uganda as relevant, (2) how increasing FP use can accelerate achievement of major health and development goals in the country, and (3) what is known from relatively recent research on the factors that influence practice of family planning in the country and justification for the paper.

9. On line 111, the authors point out the need for more recent evidence on factors associated with FP use in Uganda, but for the study they use data that is now seven years old (even though there was a DHS in Uganda in 2021). At some point in the paper (probably not intro) this will need to be explained or at least recognized as a limitation.

Materials and methods

10. Line 228 - Why the decision to measure age categorically rather than continuously in the final model? If you are able to show that positive differences in women’s age are associated with their odds of FP use when age is modelled continuously, I would do it that way.

11. Line 257-259: Did you calculate the VIFs for all variables that proved significantly associated with outcome in the bivariable analysis and then exclude those with a VIF > 4; or did you calculate the VIF first for all variables before the bivariate analysis and conduct the bivariate analysis for all covariates with a VIF < 4? Intuitively , the best way seems to be removing variables based on VIF after determining which ones are significantly associated with the outcome in bivariate tests (which is what I think the authors did but its not clear).

12. The data have a hierarchical structure (i.e., individual women are nested in households, households nested in enumeration areas, and enumeration areas nested in stratum), however, it is not clear how the authors accounted for this in the modelling strategy? Doing so is quite crucial since failing to account for clustering in analysis can result in biased inference which generally results in Type 1 error. The authors need to address this. Depending on the information available to them they can use a model- or design-based approach for this (the easier would be a model-based approach that incorporates random effects, I think)… but given the multiple levels of clustering I would imagine the authors should compare the different ways to fit the mixed-effects model and select the model with the highest “goodness of fit”.

Results of the study

13. Table 2 is a bit confusing with regards to “pattern of family planning use”. Cannot a woman have used a method before AND after last birth? Similarly, a current user could also be a woman that has used since last birth? Unless this finding is really important, you might consider removing it from the table.

14. In table 2, under current use by methods, it is written that 458 women are currently using a traditional method and below under “intention to use contraceptive” it is written that 504 women are using a traditional method. I think it might be more useful to present the breakdown of users/non-users in terms of their desire to space and/or limit future pregnancy, i.e. (1) women that desire to space or limit that are currently using a method, (2) women that do not have a birth timing preference, but are using a method anyway, (3) women that want a child soon and are using a method, (4) women with a desire to space or limit that are not using a method, (5) women that have no preference and are not using a method, (6) women that want a child soon and are not using a method.

15. Line 310 – Because the data are cross-sectional, it is hard to tell whether some of the independent variables are truly determinants, but you can call them “factors associated with family planning uptake”.

16. Lines 323-326 refer to the “goodness of fit of the model”. I think this really only makes sense if you describe what other models you fit and compared when you were determining which to report as the final model. I am unfamiliar with the use of Pearson chi-sq test for assessing goodness of fit, however with logistic mixed effects models (which is what I think the authors should use given the multi-level nature of the data), I think the authors should compare the AIC/BIC or loglikelihood values associated with models with alternative specifications for dealing with clustering and chose the most appropriate one that way. This can be described in the methods section, rather than in the results section.

17. The variable “can a woman be pregnancy after birth” needs to be described more… When after birth? Are you referring to women’s awareness of their fertility postpartum (i.e., do women know WHEN they return to being fertile after childbearing)… And what does the finding imply?

18. The authors chose to retain a lot of variables in the final model. My advice is (1) the authors should frame this by stating the hypothesis that multi-level sets of factors affect the outcome, and these levels reflect women’s individual level characteristics (e.g., age, education), household/spousal characteristics (e.g., wealth, husband education), geographic context (e.g., rural/urban, region), and their exposure to programs (e.g., heard of FP on TV, in newspaper); (2) then, the authors should fit different versions of the model that they selected based on comparing AIC and BIC values for versions that adjust for (a) individual, household and geographic covariates, and (b) individual, household, geographic and programmatic covariates. AIC and BIC values reported for each model should be included in the analysis.

Discussion and conclusion

19. Since the paper already has an introduction, I do not think that the sub-section under the discussion should also be called an introduction.

20. The findings seem to be that in 2016, women’s age, educational attainment and household wealth status were associated with their odds of currently using contraception. I think the discussion of these findings need to be framed in the context of an ongoing policy or programmatic deliberating occurring in Uganda to help readers understand why these findings are important. On the surface, these are relatively old data that tell us what we already know.

21. What do the authors think about the litany of null effects for the programmatic covariates (except “heard about FP on radio)? Based on the finding, should the discussion urge programs to do a better job at implementation? Since its not possible to ascertain whether the programmatic exposures occurred before or after the outcome of interest, does it make sense to show these results?

22. Line 410-412. The authors say that their analysis reveals that interventions that bring FP closer to women would increase uptake more than static services, but I cannot locate this finding in the analysis or discussion. I do not understand why the authors state this as an implication.

23. Lines 416-427. A major limitation is that the data are cross-sectional and the authors could not establish temporality between participants exposure to some of the independent variables (i.e., program exposure) and the outcome. The data are also 7 years old so the implications of the analysis may be too dated.

7. PLOS authors have the option to publish the peer review history of their article (what does this mean?). If published, this will include your full peer review and any attached files.

**Do you want your identity to be public for this peer review?** For information about this choice, including consent withdrawal, please see our Privacy Policy.

Reviewer #1: No

Reviewer #2: No

---

## [Editor Report · Decision Letter 2]

9 Nov 2023

Family planning uptake and its associated factors among women of reproductive age in Uganda: an insight from the Uganda Demographic and Health Survey 2016

PGPH-D-22-01403R2

Dear Mr. Ochen,

We are pleased to inform you that your manuscript 'Family planning uptake and its associated factors among women of reproductive age in Uganda: an insight from the Uganda Demographic and Health Survey 2016' has been provisionally accepted for publication in PLOS Global Public Health.

Best regards,

Claire E. von Mollendorf

Academic Editor